# Detection of spike protein in term placentas of COVID-19 vaccinated and/or SARS-CoV-2 infected women

Catharina Bartmann[1]*, Vanessa Schmidt[2], Michael Mörz[3], Michael Schwab[1], Monika Rehn[1], Bettina Blau-Schneider[1,4], Achim Wöckel[1], Ulrike Kämmerer[1]

**1** Department of Obstetrics and Gynecology, University Hospital, Würzburg, Germany, **2** Inmodia GmbH, Institute for Molecular Diagnostics, Passau, Germany, **3** Institute of Pathology 'Georg Schmorl', The Municipal Hospital Dresden-Friedrichstadt, Dresden, Germany, **4** St. Josefs-Hospital Wiesbaden, Academic Teaching Hospital, Department of Obstetrics and Gynecology, Wiesbaden, Germany

* Bartmann_C@ukw.de

## Abstract

### Introduction

COVID-19 (Corona Virus Induced Disease-19) caused by the SARS-CoV-2 corona-virus can be a serious in pregnancy. Therefore, vaccination with modRNA vaccines was recommended depending on the immunity status for women of reproductive age and pregnant women since 2022. However, there are only preliminary data on trans-placental transmission of the virus and modRNA from genetic vaccines so far.

### Methods

The study population included 106 women who have given birth at the Department of Obstetrics and Gynecology, University Hospital of Würzburg during November 2020 to October 2022. In addition to medical data and vaccination history, immunohisto-chemical examination of the placenta was performed with antibodies against SARS-CoV-2 spike and nucleocapsid proteins. RNAscope in situ Hybridization was used to show RNA detection in positive placental tissues as a proof of concept.

### Results

Altogether, 87% of participants received at least one vaccine dose against SARS-CoV-2 and 56 women (42 vaccinated, 14 not vaccinated) contracted COVID-19. In total, 31 placentas were found positive for the spike protein. Spike positive cells were predominantly Hofbauer cells and trophoblasts. In three cases of vaccinated and then infected woman, an additional nucleocapsid staining was detected, but there was no significant difference in staining pattern in correlation to the vaccine/COVID-19 status. Interestingly, we did not find viral RNA in the investigated samples, but we

**Data availability statement:** All relevant data are within the paper and its Supporting Information files.

**Funding:** The author(s) received no specific funding for this work.

**Competing interests:** The authors have declared that no competing interests exist.

**Abbreviations:** COVID-19, CoronaVirus Induced Disease-19; WHO, World Health Organization; CTB, cytotrophoblast; STB, syncytiotrophoblast; NICU, neonatal intensive care unit; RT-PCR, reverse-transcriptase polymerase chain reaction; RT, room temperature; GeZeCO, "*Geburt in Zeiten von COVID-19" (*delivery in times of COVID-19); EMA, European Medical Agency; DW, destilled water; PBS, phosphate buffered saline; APC, antigen presenting cells; Fig., Figure; HE, Hematoxylin and eosin; DAB, diaminobenzidine.

could show a positive in situ Hybridization of BNT162b2 and S-encoding mRNA-1273 in two individual samples.

## Discussion

The spike protein of SARS-CoV-2 has been be detected in placental Hofbauer and Trophoblast cells as well as villous endothelia after infection and vaccination indicating a possible transplacental transfer or uptake. These findings may suggest a potential for transplacental transfer or cellular uptake; however, the extent, mechanisms, and clinical significance of this phenomenon remain to be fully understood.

Clinical trial registration: DRKS00022506.

## 1. Introduction

A COVID-19 (Corona Virus Induced Disease-19) pandemic was declared by the WHO in March 2020 [1] and was caused by the SARS-CoV-2 coronavirus [2]. Symptoms were predominantly mild with fever, cough, sore throat, headache, rhinorrhea and myalgia [3, 4]. However, under certain circumstances, a secondary acute onset of severe courses (hyper-allergic reactions; [5–7]) with necessary intensive medical treatment and even death was observed [8].

In a meta-analysis, most common symptoms in pregnant women were mild with fever, cough and dyspnoea [9, 10]. Rare pathologic effects were seen in placentas from COVID-19 diagnosed mothers compared to negative controls [11]. A review summarizing the clinical course of COVID-19 in pregnancy during 2020 concluded, that the maternal risk of severe COVID-19 during pregnancy was not greater than for the general population and neonatal outcome was not influenced by the disease [12]. However, an increased risk for severe illness which leads more frequently to intensive care stays and ECMO [extracorporeal membrane oxygenation) therapy in pregnant women was seen in another data collection [13–15]. Furthermore, pregnant women with COVID-19 seem to have a higher risk of pregnancy disorders such as pre-eclampsia [16]. From the fetal side, more premature births and neonatal morbidities were reported in 2021 [17] and it was described later that in some cases, COVID-19 could cause endothelial damage and a higher risk of thrombose and embolism [18].

The placenta plays a crucial role as a barrier for transmissions of maternal infections to the fetus. On the fetal side, it consists of chorionic villi, formed by cytotrophoblast cells (CTB) and the syncytiotrophoblast (STB) [19]. Within the chorionic villi there is a specific form of immature macrophages of fetal origin, the Hofbauer cells [20].

The transmission probability of SARS-CoV-2 from mother to fetus is very low [21–23], and in the majority of cases of RT-PCR identified SARS-CoV-2 infected pregnant women, placentas were RT-PCR negative [24]. However, in three cases of RT-PCR positive placentas derived from pregnant women with SARS-CoV-2 RT-PCR positive nasal swabs (two without any symptoms and one with mild COVID-19

symptoms), RNA of SARS-CoV-2 was confirmed in placental tissue with the use of antisense-RNA probes, but only in one of three cases hybridisation with the sense probe indicated viral replication in the STB at low levels [25]. Other investigations showed virus-RNA and/or spike protein in the placenta, but the impact on the fetal compartment seemed to be negligible [26–30].

Various vaccines against the SARS-CoV-2 virus were developed with the focus on the spike protein of SARS-CoV-2 as antigen in a very short time. Available since December 2020, the main principle of the new type of genetic vaccines (definition according to the Paul Ehrlich Institute) used in Germany was a combination of base-modified mRNA (modRNA) within a lipid nanoparticle envelope [31], here Comirnaty (Pfizer/BioNTech) and Spikevax (Moderna).

Although the genetic COVID-19 vaccines were not fully approved for pregnant women during our trial, modRNA vaccination was recommended by the EMA at the beginning of 2022 [32] and even earlier, starting September 2021 by the German "Ständige Impfkommission (STIKO)" [33], for women in the second trimester of pregnancy. This recommendation was based on the idea that pregnancy itself was a risk factor for a severe course [34]. Further, there was evidence for transplacental transfer of vaccine-induced anti spike antibodies [35, 36] which was assumed to have a protective effect for the newborn. No official data from registry studies suggest a disadvantage of vaccination in pregnancy at those times [37] which was later confirmed in a meta-analysis [38]. As a result and due to intensive marketing campaigns and several restrictions in daily life including access to hospitals and delivery rooms for persons who were not vaccinated against SARS-CoV-2 [39] vaccination rates of pregnant and breastfeeding women increased significantly, but did not reach the percentages of the general population [40]. This was despite the fact that there were no data available at this time about a possible transfer of the vaccine or spike protein through the placenta. After the recruitment of out participants, in an in vitro model, vaccine derived modRNA was taken up in placental tissue albeit without detectable effect [41].

A more recent review by Zhong C et al 2024 [42] summarized the limited data at that time on *in vivo* vaccine RNA transfer into the placenta. In a mouse model, Chen and colleagues demonstrated, that mRNA-1273 can cross the placenta within one hour after intramuscular injection into the pregnant mice [43]. In a research letter from the group of Monica Hannah [44], two cases were reported in which vaccine RNA was detected in the villi of human placentas. Here, both mothers had received their last of 2–4 RNA-vaccinations shortly before delivery (2 or 10 days respectively). The RNA was detectable via PCR and RNAScope-based in situ hybridisation, while spike protein was detectable by Western Blot in only one of the placentas.

In our study, we were interested if the spike protein could be found in placentas from women who had either suffered from COVID-19 during pregnancy or had been COVID-19 vaccinated or both by immunohistochemistry. Further, in positive cases, if the spike protein was located on the maternal or fetal side – the latter confirmed that the spike protein and/or the genetic information of the virus or vaccines were able to pass the maternal-fetal border. And finally, it was of interest if the modRNA derived from the vaccines was able to reach the placental compartment in our cohort and to correlate with spike protein -positive fetal cells in placental tissues.

## 2. Materials and methods

### 2.1. Description of the study participants

The study population investigated herein was part of the GeZeCO trial [45] and included women who had given birth at the Department of Obstetrics and Gynecology, University Hospital of Würzburg from November 2020 to October 2022. The study adhered to the Declaration of Helsinki and was approved by the ethics committee of the University of Würzburg (No. 70/20 Amendment). All participants received verbal and written information and agreed to participate with written informed consent. Here we used the following inclusion criteria: sufficient placental tissue available from women with either documented symptomatic COVID-19 or who received at least one shot of the COVID-19 vaccine with or without subsequent COVID-19.

Of the 106 cases analyzed here, 14 were not vaccinated against COVID-19 but diagnosed with that disease by symptoms and SARS-CoV-2 genome RT-PCR test. All others received at least one injection with a COVID-19 vaccine (see Table 1) and had subsequently either a symptomatic SARS-CoV-2 RT-PCR-confirmed diagnosis (n = 42) or RT-PCR test negative symptomatic respiratory diseases (n = 20). The remaining vaccinated participants showed no signs of any infection (n = 30) during pregnancy. Pregnant women without any vaccination against SARS-CoV-2 and not diagnosed with COVID-19 or insufficient placenta samples were excluded from this analysis.

Women included in the trial gave birth from the 35th until the 42nd week of pregnancy. In total, there were 101 singleton and 5 twin pregnancies. With regard to the neonatal outcomes and placental analyses, the first baby was taken into account only. The mean neonatal birth weight was 3441.0g (±513.8g [SD]). The minimum/maximum APGAR values were 1/10 (mean 8.89 ± 1.29) after one minute, 6/10 (mean 9.66 ± 0.77) after 5 minutes and 8/10 (mean 9.85 ± 0.45) after 10 minutes. The pH value of the umbilical artery was 7.26 (±0.06), the base excess (BE) came to −4.55 (±2.77) mmol/l. The pH value of umbilical vein was 7.35 (±0.06) (Table 2).

## 2.2. Sample collection and immunohistochemistry

Placental samples were collected directly after delivery. Cubes of 1 $cm^3$ each were cut under sterile conditions from the central and peripheral parts of the placenta, packed in sterile micro-tubes, snap frozen in liquid nitrogen and stored at −80°C.

Because antibodies against the spike (rabbit polyclonal, ProSci #9083) and nucleocapsid protein (mouse monoclonal, Pros-Sci #35−720) of SARS-CoV-2 as well as in-situ hybridization probes (see 2.3.) were specified for paraffin sections, tissue samples were transferred to paraffin. For this, frozen tissues were equilibrated to −20°C over night, defrosted for 30 min to 4°C, subsequently fixed in 8% formalin (buffered in PBS/Dulbeccos) for 24 h and finally embedded into paraffin blocks via routine automated dehydration and embedding procedure at the Institute of Pathology of the University of Würzburg.

Paraffin blocks were cut into 2 μm sections, mounted on glass-slides (Superfrost, Langenbrink, Emmendingen, Germany) and dried overnight at room temperature (RT). Sections were dewaxed twice with xylene and rehydrated in a graded series of ethanol and distilled water (DW). For antigen demasking, slides were pretreated in a 10 mM sodium citrate buffer (pH 6.0) for 3x5 minutes (microwave oven; 750 W/s) and then thoroughly washed in DW. Endogenous

**Table 1. Substances used for the vaccine injections in our cohort in relation to the occurrence of COVID-19 during pregnancy.**

| Kind of vaccine (total number of injections) | | COVID-19 during pregnancy | |
|---|---|---|---|
| | | No | Yes |
| 1st injection (n = 92) | Comirnaty (n = 71) | 42 | 29 |
| | Spikevax (n = 11) | 4 | 7 |
| | Vaxzevria (n = 9) | 4 | 5 |
| | Janssen (n = 1) | 0 | 1 |
| | | | |
| 2nd injection (n = 83) | Comirnaty (n = 74) | 43 | 31 |
| | Spikevax (n = 8) | 3 | 5 |
| | Vaxzevria (n = 1) | 0 | 1 |
| | | | |
| 3rd injection (n = 49) | Comirnaty (n = 49) | 24 | 25 |
| 4th injection (n = 1) | Comirnaty (n = 1) | 0 | 1 |

**Table 2. Obstetrical and neonatal characteristics in relation to vaccination and/or COVID-19.**

| Parameter | COVID-19 only (n = 14) | | Vaccinated, no disease (n = 30) | | Vaccinated and COVID-19 (n = 42) | | Vaccinated and respiratory diseases (n = 20) | | P- value* |
|---|---|---|---|---|---|---|---|---|---|
| | Mean | SD | Mean | SD | Mean | SD | Mean | SD | |
| Week of pregnancy | 39.8 | 1.8 | 39.7 | 1.4 | 40.1 | 1.6 | 39.7 | 1.5 | 0.60 |
| Birth weight (kg) | 3.465 | 0.528 | 3.289 | 0.496 | 3.503 | 0.475 | 3.523 | 0.595 | 0.28 |
| APGAR (1 min) | 8.71 | 0.73 | 9.10 | 0.92 | 8.88 | 1.45 | 8.70 | 1.69 | 0.23 |
| APGAR (5 min) | 9.43 | 0.94 | 9.77 | 0.57 | 9.69 | 0.75 | 9.60 | 0.94 | 0.54 |
| APGAR (10 min) | 9.86 | 0.36 | 9.83 | 0.46 | 9.86 | 0.47 | 9.85 | 0.49 | 0.96 |
| pH of umbilical artery | 7.25 | 0.06 | 7.27 | 0.05 | 7.26 | 0.07 | 7.28 | 0.06 | 0.35 |
| pH of umbilical vein | 7.33 | 0.07 | 7.35 | 0.05 | 7.34 | 0.06 | 7.37 | 0.05 | 0.313 |
| Base excess (mmol/l) | −5.12 | 3.14 | −4.10 | 2.64 | −5.22 | 2.42 | −3.38 | 3.10 | 0.053 |

* p-value via Kruskal-Wallis Test

peroxidase activity was blocked with 3% hydrogen peroxide in methanol for 10 minutes at RT, slides then washed in DW and equilibrated to pH7.2 with PBS/Dulbeccos. Possible unspecific antibody binding of the tissue was blocked with FC blocking reagent (innovex biosciences) for 20 minutes at RT prior to the application of specific antibodies. Primary antibodies for immunohistochemistry were summarized in Table 3.

The sections were incubated overnight at 4°C with the respective primary antibodies or unspecific negative control antibodies (both from DAKO) diluted in "diluent reagent" (DAKO), washed with PBS and incubated with the corresponding secondary antibody (anti-Mouse/HRP or anti-Rabbit/HRP, DAKO, ready to use) for 30 min at RT. Peroxidase activity was developed with the DAB(diaminobenzidine)plus substrate kit (DAKO) for 5–10 min under microscopic control, resulting in brown staining. For double immunohistochemistry on the Ventana Discovery XT slide staining instrument (Roche Ventana), spike-positive sections were washed with the Ventana Reaction buffer followed by incubation with the Anti-CD68 antibody for 30 min at RT in a humidified atmosphere. Then, slides were incubated for 30 min at RT with the Zytochem Plus (AP) polymer anti-mouse antibody (ZUC077, Zytomed) and washed with the Ventana reaction buffer. AP-red (ZUC001, Zytomed) was developed for 10 min according to the supplier's protocol, sections then thoroughly washed in destilled water. Tissues were counterstained with hematoxylin and eosin (HE), fixed in graded alcohol till xylene, embedded with Vitroculd (Langenbrink) and analyzed using a light microscope (Othoplan, Leica, Germany).

Positive control samples were stained in parallel as described in detail in [46]. Placenta tissues collected before 2018 served as negative control samples. Histology and staining were evaluated by two independent observers (UK and MM).

**Table 3. Primary antibodies used for immunohistochemistry.**

| Target Antigen | Supplier | Clone /Nr. | Species | Dilution |
|---|---|---|---|---|
| SARS-CoV-2 Spike subunit 1 | ProSci | 9083 | Rabbit | 1:500 |
| SARS-CoV-2 nucleocapsid | ProSci | 35-720 | Mouse | 1:500 |
| CD68 (monocyte marker) | DAKO | PG-M1 | Mouse | 1:100 |
| Negative control mouse | DAKO | GA750 | Mouse | Ready to use |
| Negative control rabbit | DAKO | GA600 | Rabbit | Ready to use |

## 2.3. RNAscope assays

*In situ* detection of mRNA derived from either the SARS-CoV-2 virus or from the RNA vaccines on selected paraffin tissue sections shown to be positive via immunohistochemistry was performed with the RNAscope assay following the manufacturer's kits instructions (ACD, Bio-Techne Ltd, Minneapolis, MN, USA). Here, specific probes for SARS-CoV-2 Wuhan strain spike RNA sequence (RNAscope Probe V-nCoV2019-S Wuhan #848561-C1) or Pfizer/Biontech Comirnaty (RNAscope Probe BNT162b2-C1 #1104241-C1) or for Spikevax (RNAscope Probe S-encoding-mRNA-1273-C1 #1104251-C1) were applied on consecutive sections of placental tissue. Detection of the amplified RNA was performed with the RNAscope® 2.5 High Definition(HD) – RED Assay #322350s. Nuclei were counterstained with hematoxylin and the images were acquired by light microscope (Leica, Germany).

## 2.4. Data analysis and statistics

The values are presented as absolute numbers or percentages as well as means with standard deviations (SD). The software IBM SPSS Statistics 28.0 was used to create tables and perform statistical analyses. Mann-Whitney-U test, Kruskal-Wallis-test and Pearson's chi-squared test were performed and p-values ≤0.05 were considered statistically significant. GraphPad Prism 9.5.1. was used for graphs.

## 3. Results

### 3.1. COVID-19 vaccine status

Altogether, 92 of the 106 (87%) women received at least one dose of vaccine against SARS-CoV-2. Of those, one woman reported four shots with Comirnaty (2 before pregnancy, 2 in the second trimester), 49 had three injections (2 basic and booster) with nine completing the booster before pregnancy, five in the first, 30 in the second and five in the third trimester. Of the 33 women with two injections, nine received the second shot in the third, seven in the second and three in the first trimester. One woman did not specify the date of the second shot and 13 had both shots before pregnancy. Of the nine who had one injection only, six had been vaccinated before pregnancy, two in the first and one in the third trimester. The substances used for vaccination are summarized in Table 1 (and supplemental S1 Table).

### 3.2. Obstetrical characteristics in relation to vaccination and/or COVID-19 status

Of the 92 vaccinated women, 42 (45.6%) suffered from consecutive COVID-19 and were proven SARS-CoV-2 positive via RT-PCR testing, four (4.3%) reported typical symptoms of COVID-19 without positive tests, 16 (17.4%) reported a common cold and 30 (32.6%) had no symptomatic infectious diseases during pregnancy. There were no differences in any of the obstetric clinical data described above (compare Table 2).

### 3.3. COVID-19 and disease symptoms in pregnancy

Of the 56 patients (42 vaccinated, 14 not vaccinated) who suffered from COVID-19) during pregnancy, five (8.9%) were infected in the first trimester, 22 (39.3%) in the second trimester and 24 (42.8%) in the third trimester. Three (5.3%) patients did not provide information on the gestational stage of their COVID-19 disease. Two anti-SARS-CoV-2 vaccinated women were diagnosed two times with COVID-19 in both the second and third trimester. All patients were asked about the typical COVID-19 symptoms in a questionnaire; the results are summarized in Fig 1 and 2.

With one exception, there were no significant differences in most of the reported symptoms and disease burdens between vaccinated versus non-vaccinated women. Without vaccinations, 8 out of 14 (57%) and with vaccinations 10 out of 42 women (24%) reported taste and odor disorders (p = 0.012). The majority (n = 50) had nasal congestions followed by sore throats (n = 45) and coughing (n = 44). 26 women reported shortness of breath. Fever (n = 25) and

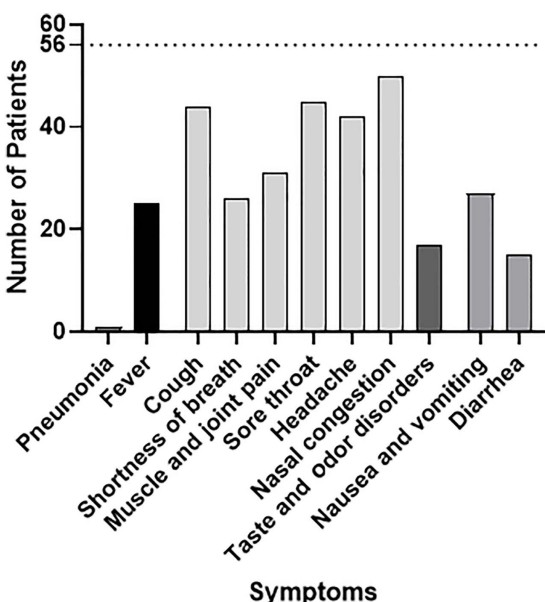

**Fig 1. Disease symptoms of COVID-19 affected women (n = 56; 14 non vaccinated, 42 vaccinated), predominantly symptoms common for respiratory virus infections.**

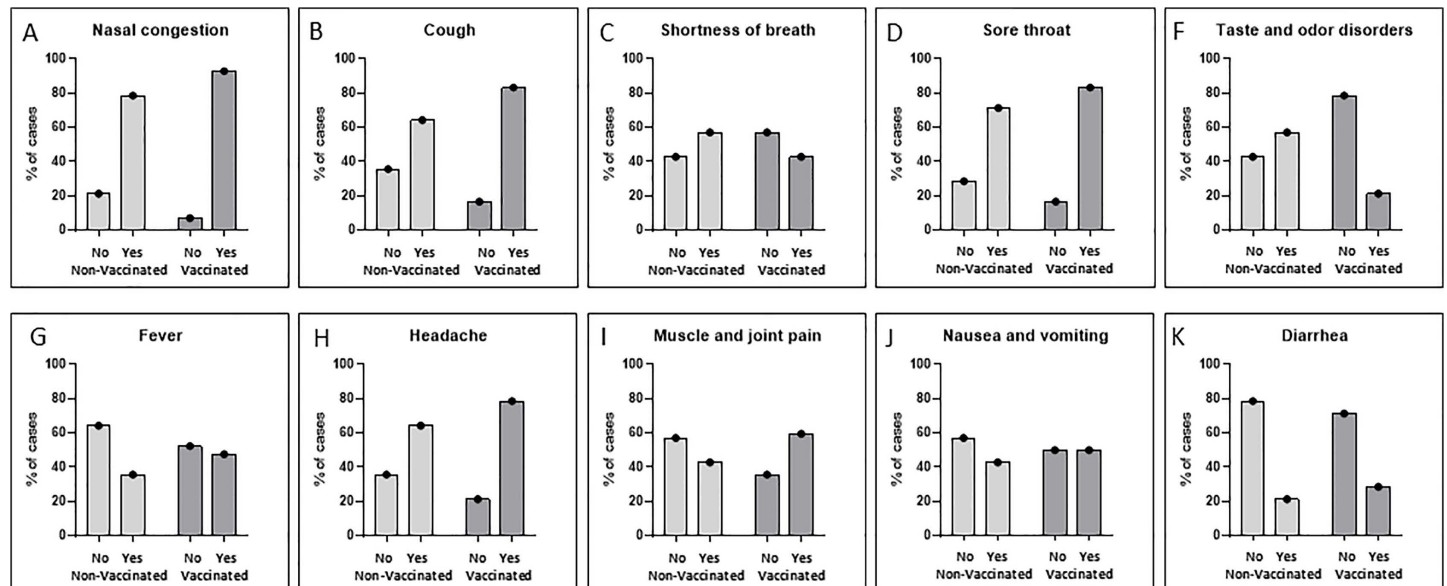

**Fig 2. Relative distribution of disease symptoms of COVID-19 affected women between the different groups of non-vaccinated (n = 14) and vaccinated (n = 42) women.** By chi-square testing, only taste and odor disorders (F) differed significantly (p = 0.012) in that the non-vaccinated women reported this symptom more frequent.

headache (n = 42) were similar between groups, while muscle- and joint pain affected relatively more women (n = 31) from the vaccinated group. Several women reported gastrointestinal problems with nausea and vomiting (n = 27) and/or diarrhea (n = 15).

Only one woman had a more severe disease with hospitalization due to pneumonia in late pregnancy (not included in the graph). She reported two vaccinations with BNT162b (Comirnaty) before pregnancy and a third dose in the second trimester.

### 3.4. Neonatal outcome

Of the 106 neonates counted (in the case of twins, the first newborn), in total eight had malformations or congenital diseases. Three of those diagnoses were not relevant for the newborn in the postpartum period, so that no admission to the neonatal intensive care unit (NICU) was arranged. Five neonates were subjected to NICU treatment with malformations or congenital diseases and six other neonates were admitted to the NICU due to respiratory adaptive problems, maternal mental health medication or prematurity. According to the medical documentation, no newborn of this study was admitted to the NICU due to a maternal COVID-19 infection. As shown in Table 4, there were no significant differences of the diagnosis of neonatal diseases/congenital malformations and of the admission to the NICU between the neonates of the COVID-19 group and the three subgroups of vaccinated mothers.

### 3.5. Immunohistochemistry

In total, 31 placentas were found positive for the spike protein in at least one cell type (compare supplemental S1 Table). Three of them were from non-vaccinated women which had suffered from symptomatic and tested positive forSARS-CoV-2 during pregnancy (2x second and 1x third trimester). Spike protein-positive placental tissues were from vaccinated women who either reported no infections (n = 11) or tested positive symptomatic COVID-19 (n = 12) or an unspecific respiratory disease (n = 5) during pregnancy.

In three cases, parallel to spike protein-staining (Fig 3F-H) additional nucleocapsid protein staining was detected (Fig 3B-D). All three spike and nucleocapsid protein positive samples were from women who had COVID-19 after vaccination in the last weeks of pregnancy (gestational ages: 36 and 37). In one case, nucleocapsid protein was found in the STB only and in the other two samples in addition in cells in the intervillous space, thus resembling leukocytes of the mother. Spike protein positive cells on the fetal side of the placenta were predominantly Hofbauer cells (Fig 4A,B; 24 cases) which was confirmed by doublestaining against the monocyte-specific marker CD86 (Fig 5) followed by STB (Fig 3F-H; +Fig 5; 19 cases), the trophoblast layer (Fig 4C,D; 11 cases) and endothelia of villous vessels (Fig 4E,F; 9 cases) or combinations of those cells. On the maternal side, the intervillous immune cells were positive in two cases only (Fig 4).

The distribution of spike protein positive cells in correlation to the vaccine/COVID-19 status is summarized in Table 5. There was no significant difference in staining patterns amongst the subgroups.

Additional staining was performed by RNAscope in individual samples (n = 9) in which we obtained a positive staining for the spike protein. Of these samples, three were nucleocapsid protein positive as well. Altogether, we detected no presence of virus derived RNA (negative for the V-nCOV2019-S Wuhan probe which we rated as negative control therefore) in the tested samples, while two positive mRNA detections with the vaccine-specific RNA probe showed the presence of

**Table 4. Neonatal outcome in correlation to the COVID-19/vaccine status of the mother.**

| | | COVID-19 (n = 14) | Vaccinated – no disease (n = 30) | Vaccinated + COVID-19 (n = 42) | Vaccinated + respiratory disease (n = 20) | p (chi-square) |
|---|---|---|---|---|---|---|
| Neonatal diseases/mal-formations | No | 13 | 28 | 38 | 19 | 0.928 |
| | Yes | 1 | 2 | 4 | 1 | |
| Admission to NICU | No | 12 | 26 | 38 | 19 | 0.761 |
| | Yes | 2 | 4 | 4 | 1 | |

NICU: neonatal intensive care unit

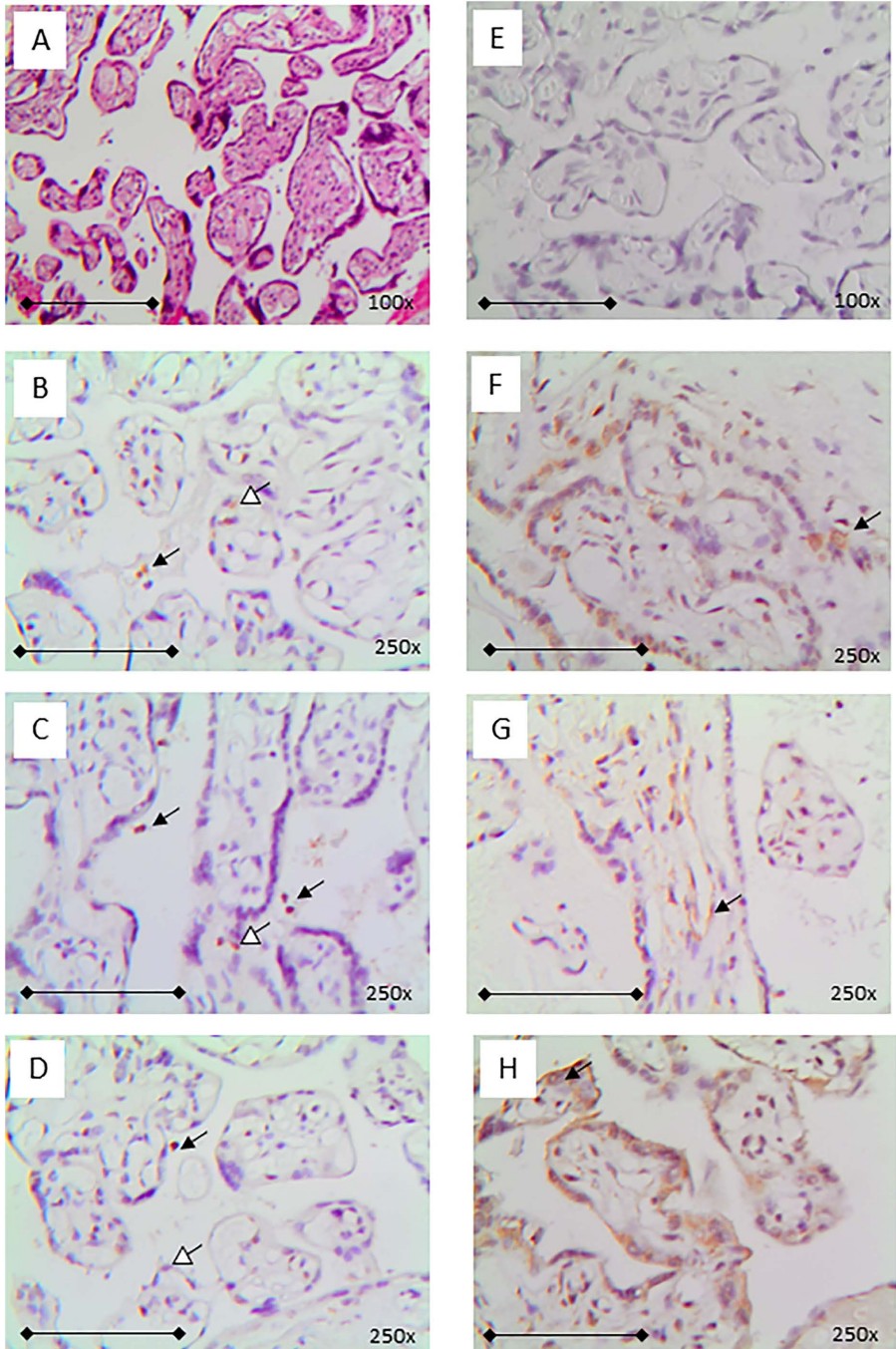

**Fig 3. Typical immunohistochemistry results on placental samples.** A: Hematoxylin and eosin (HE) staining and E: negative control immunohisto-chemistry was performed in all cases. Representative figures from the three placentas with positive nucleocapsid staining (B-D) and the corresponding spike protein positive findings (F-H). In those samples, spike protein positivity was seen in syncytiotrophoblast (STB) cells predominantly, but also in Hof-bauer cells (F, H; arrow) and endothelial cells (G; arrow. A+E: magnification x100 (bar represents 1 mm); all others: magnification x250 (bar represents 0,5 mm); counterstain HE, brown DAB+ indicates positive antibody binding. Black arrow points to intervillous stained cells, white head arrow to STB cells positive for nucleocapsid.

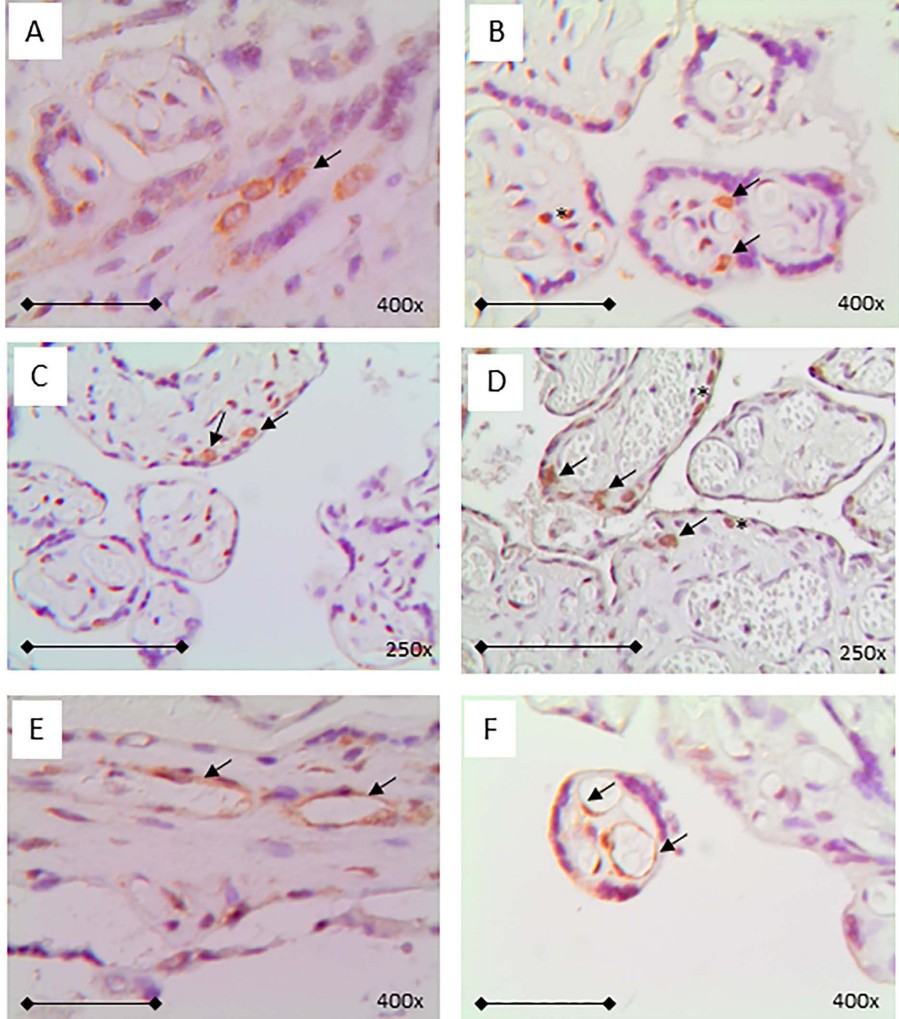

**Fig 4. Typical examples of immunohistochemical detection of spike protein positive cells at the main non-STB cell types.** A+B: Hofbauer cells were seen positive for spike protein, C+D: trophoblast layer and E+F endothelial cells of villous vessels A, B, E, F: magnification x400 (bar represents 0,25 mm); C, D: magnification x250 (bar represents 0,5 mm). Counterstain HE.

vaccine derived RNA in placental cells. In one placenta, we analyzed positive decidual surface cells for BNT162b2 (Fig 6A, B). The woman was vaccinated with Comirnaty two times before pregnancy and the third time in the second trimester. She suffered from COVID-19 in the 36th week of pregnancy. In the other placenta, we detected positive villous endothelial cells for S-encoding-mRNA-1273 (Fig 6C, D); this tissue was from a woman who was vaccinated with two injections of Spikevax before pregnancy (Fig 6C, D ).

## 4. Discussion

Many infection control measures including the development of vaccines during the COVID-19 pandemic were taken in Germany following European guidelines [47]. A new type of mRNA based genetic COVID-19 vaccines like BNT162B2 or Spikevax quickly obtained emergency use approval at the end of 2020 [48, 49] and were administed to pregnant women within a short time. This was inconsistent with the general obstetrical practice to take special care in the treatment of

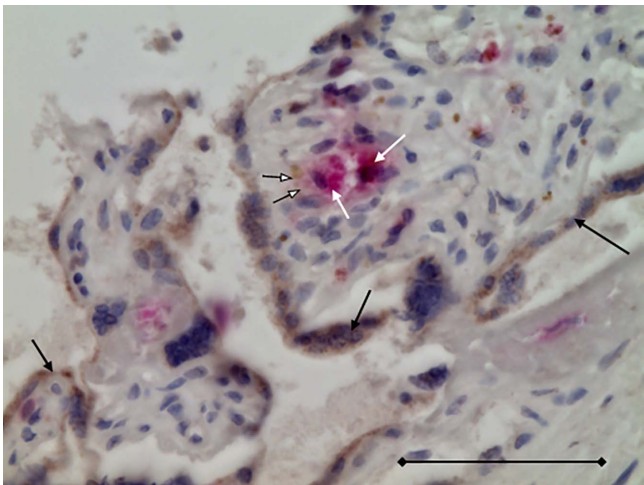

**Fig 5. Double immunostaining with anti-spike (DAB, brown) and anti-CD68 (AP, red) antibodies.** The Hofbauer cell (white arrow), clearly stained with the anti-CD68 antibody, also showed positive anti-spike staining (white arrowhead), similar to the syncytiotrophoblast cells (black arrows). Sections were counterstained with HE; magnification ×400; scale bar = 250 μm.

**Table 5. Type of positive cells in the placental samples and number of individual tissues with positive detection.**

|  |  | COVID-19 (n = 14) | Vaccine – no disease (n = 30) | Vaccine + COVID-19 (n = 42) | Vaccine + respirat. disease (n = 20) | p (chi-square) |
|---|---|---|---|---|---|---|
| **Spike** | Positive samples | 3 (21%) | 11 (37%) | 12 (28%) | 5 (25%) |  |
| Hofbauer cells | Negative | 12 | 21 | 33 | 16 | 0.659 |
|  | Positive | 2 | 9 | 9 | 4 |  |
| STB | Negative | 12 | 23 | 36 | 16 | 0.570 |
|  | Positive | 2 | 7 | 6 | 4 |  |
| CTB | Negative | 14 | 26 | 37 | 18 | 0.232 |
|  | Positive | 0 | 4 | 5 | 2 |  |
| Villous endothelia | Negative | 14 | 25 | 39 | 19 | 0.232 |
|  | Positive | 0 | 5 | 3 | 1 |  |
| **Nucleocapsid** | Positive samples | 0 | 0 | 3 (7.1%) | 0 |  |
| STB | Negative | 14 | 30 | 41 | 20 | 0.673 |
|  | Positive | 0 | 0 | 2 | 0 |  |
| Intravillous cells | Negative | 14 | 30 | 40 | 20 | 0.376 |
|  | Positive | 0 | 0 | 2 | 0 |  |

pregnant and breastfeeding women to prevent the fetus or infant from suffering adverse effects [50]. As there are still many gaps in the knowledge on COVID-19 and/or COVID-19 vaccination in pregnancy, especially about the effects on the placenta, we accessed material from an already existing sample collection (GeZeCO, Clinical trial registration DRKS00022506, compare [45, 51]) for retrospective analysis.

By the end of June 2022, about 80.6% of the German population had received at least one dose of a COVID-19 vaccine [52]. Although there was an official recommendation for pregnant women [40], the vaccination rate (approx. 60%) was significantly lower in this group compared to the general German population. Due to the inclusion criteria, our study group was either vaccinated or infected or both. From the 92 vaccinated women in our study, 90.2%

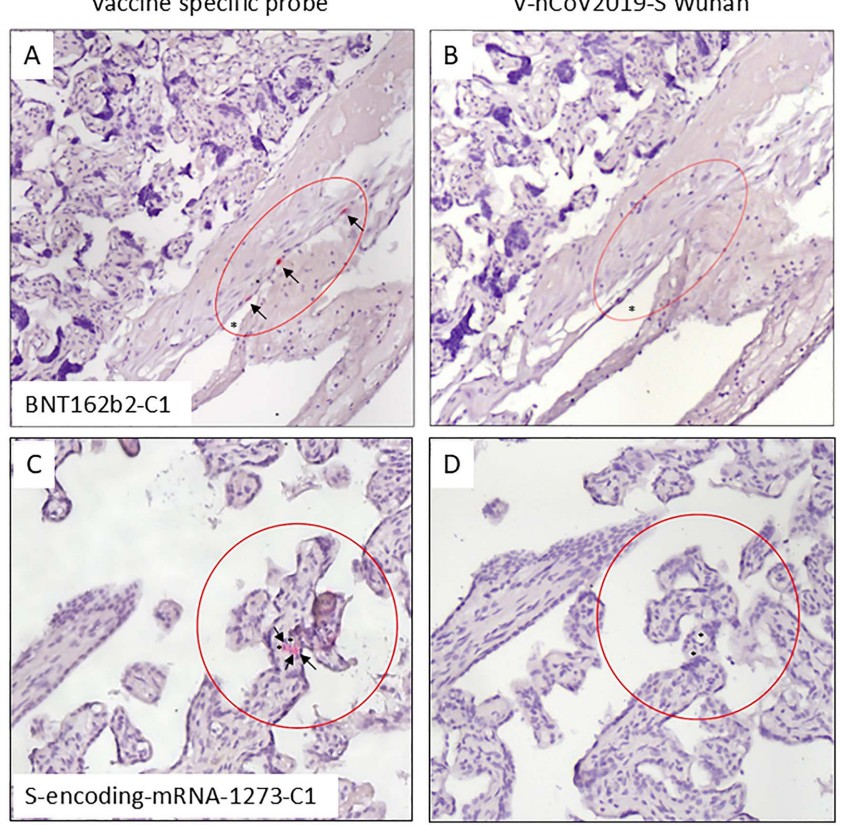

**Fig 6. RNAscope in situ Hybridization with the sensors BNT162b2 (A), V-nCOV2019-S Wuhan (B &D) and S-encoding-mRNA-1273 (C).** A, B: Placental specimen of a woman vaccinated by Comirnaty before and during pregnancy and with COVID-19 in the 36th week of pregnancy. C, D: Placental specimen of a woman vaccinated by Spikevax before pregnancy. Circles = area of interest; arrows = positive stained cells.

received a second and 54.3% a third dose of a COVID-19 vaccine. One woman had 4 doses. This acceptance of the second, third and fourth shot is clearly lower than in the general population [52] and was arguably caused by the fear of negative side effects and of harming the fetus [40, 53]. Probably due to the rather low number of cases and certainly due to our inclusion criteria (uncomplicated birth at term), we did not notice any significant differences between vaccinated and unvaccinated women related to routine obstetric features or infant outcome at birth. This must be considered in view of the results summarized in meta-analyses, in which fewer birth outcome risks (as risk of preterm delivery, small-for-gestational age in term babies and stillbirth) were reported [54] for vaccinated women but a higher risk of a caesarean section [55].

The pregnant women in our study reported typical COVID-19 symptoms like fever, coughing, sore throat, headache, rhinorrhea and muscle and joint pain, which matches the findings summarized in a review [10] and do not differ from the symptoms of non-pregnant women [3, 9, 56, 57]. COVID-19 symptoms were expected to be milder in vaccinated persons due to the postulated immune protection. This was however not seen in our trial, presumably due to the general mild course of the disease and a missing quantitative assessment of the individual symptoms in our cohort. The only significant difference was a lower rate of taste and odor disorders, rated as typical symptoms of the SARS-CoV-2 infection [8], in vaccinated women. However, COVID-19 symptoms also depend on the respective virus variant in pregnant women [58] and the vaccinated participants were infected predominantly with later variants as compared to the non-vaccinated.

Surprisingly, we identified a relatively high percentage (29%) of placenta tissues (31 cases) which were positive for the spike protein in the fetal compartment with three cases showing additional nucleocapsid protein staining on the maternal (intervillous) site of the placenta. The majority of spike protein positive placentas were from women suffering from COVID-19 during pregnancy, independent of their vaccination status. This could be caused by transmissibility or uptake of either the virus or the vaccine and a subsequent local spike production within the placenta. Alternatively, the finding could be caused by the uptake of soluble spike protein diffusing from the maternal circulation into the fetal side and entering fetal cells via phagocytosis. However, because this is a descriptive study on fixed tissue, we cannot draw conclusions about the underlying physiological mechanisms or the clinical significance of this phenomenon.

In the spike protein positive cases a negative impact on the placenta or the fetus was not observed. This could lead to the assumption, that the placenta – especially STB and Hofbauer cells – was able to trap the virus/ vaccine/spike-protein without transmitting it to the fetus or reacting with a clear inflammatory response. This corresponds to the data summarized by Li and coworkers which showed only rare impact on placental integrity in pregnant women suffering from COVID-19 [34]. As seen previously [26, 29, 59, 60], spike protein was found in the STB in 61% of positive placentas. The STB is in direct contact with the maternal blood and expresses the main receptor for the spike protein, ACE2 which was shown to mediate the STB infection [59, 61]. In addition, STB express the second important receptor for this SARS-CoV-2 spike protein, Neuropilin-1 [62–64], so it is very likely that those cells take up any circulating virus or spike protein (derived from the vaccines and present in the blood of vaccinated persons via exosomes [65]). Another possible mechanism could be the crossing of the lipid-nanoparticles carrying the modRNA, which are highly stable in the blood [66] from the maternal side resulting in an effective transfection of the placental cells which then produce the spike protein based on the coded sequence. The ACE2 receptor was also found in the CTB [59] which also express Neuropilin-1 [62]. Accordingly, in 11 cases, spike protein was also found in those cell types in our samples, however, in placentas from vaccinated women only, equally distributed in tissues from women either with or without accompanying disease.

In 77% of spike protein positive tissues, a clear spike protein expression was seen in the Hofbauer cells. This corresponds with the detection of viral RNA in those cells in a human placental infection model [67] and to other *in vivo* studies [68]. Spike-protein positive Hofbauer cells were predominantly seen in placentas of vaccinated women, and only in two cases (6%) of non-vaccinated COVID-19 infected women. Thus, the lack of detection of SARS-CoV-2 components in Hofbauer cells published for a placenta from a case with transplacental infection of a neonate with SARS-CoV-2 woman may fit with this rare expression in those infection-only cases [27]. The relatively large percentage of spike protein positive Hofbauer cells in vaccinated women might reflect the fact, that those immune cells collect foreign proteins like the spike-proteins in their function as antigen presenting cells (APC). Further in case of modRNA vaccines, the declared target of the lipid nanoparticles carrying the modRNA was the dendritic cells in the lymphoid organs [69, 70], an APC type, like the Hofbauer cells [71]. Accordingly, all but two of the positive cases were injected at least once with Comirnaty, one case with Spikevax and only one with Janssen. So, fetal immune cells might be directly impacted by circulating lipid nanoparticles from the maternal vaccination – which circulate in the maternal blood for at least 28 days post vaccination [72]. However, due to the low number of samples and to the descriptive character of our investigation only, we can only speculate about those possible pathophysiological mechanisms.

Additional positive nucleocapsid protein staining was obtained in three tissues from vaccinated and subsequently COVID-19 infected women. Two contracted the disease 2−3 weeks before giving birth and thus representing a fresh infection and one nine weeks before delivery showing a remarkably long persistence of nucleocapsid protein in the placenta. In a Swiss study, the ability of the virus to replicate in the placenta was observed [73]. This could explain the long observed period of virus nucleocapsid protein persistence. It appears that viral replication in the placenta correlates with a negative outcome in terms of stillbirth [73]. In comparison with other studies [74, 75], we detected nucleocapsid protein positive placenta specimen only in 9.6% of the women reporting COVID-19. This could be due to the fact that we had – with one exception – only mild cases of COVID-19 among our study patients. Further, this corresponds to a study from New York

that observed no nucleocapsid protein staining in 64 placentas from women with COVID-19 [76]. Since we detected the nucleocapsid protein only in maternal cells and not in fetal cells, we could speculate that the spike protein found on the fetal side is not derived from viral infection. Instead, it is more likely the result of either uptake of freely circulating spike subunit-1 protein or local spike production by vaccine-transfected cells. The maternal immune cells expressing the nucleocapsid protein most likely acquired it through the uptake of infected and subsequently lysed cells. Because nucleocapsid protein is the most abundant structural protein of coronaviruses [77], the absence of detectable S1 spike subunit in maternal placental cells may simply reflect that the amount of internalized spike protein was below the detection threshold. To further investigate a potential local source of the observed spike protein, we analyzed the three nucleocapsid-positive samples and six additional samples using the mRNAscope technique.

A positive finding for the virus-RNA probe would have allowed to rate an active infection of the placenta and positive finding for vaccine specific probes could allow to discuss the transplacental transmission of the genetic vaccines. Here, all samples were negative for the mRNA of the virus as seen before by Santos and coworkers [78]. However, in two samples we found a specific pronounced hybridization of vaccine specific RNA probes indicating the existence of traces of the vaccine: The RNA representing BNT162b2 in a sample of a woman who had been vaccinated by Comirnaty three times before and during pregnancy and had contracted COVID-19 in the 36th week of pregnancy was seen in cells of the decidual surface, representing the border between the maternal and fetal compartment. Surprisingly the Spikevax identifying S-encoding-mRNA-1273 was positive in endothelial cells of villous capillaries of a placenta derived from a woman who had been vaccinated two times with Spikevax before pregnancy. These observations are consistent with the present work by Gonzalez and coauthors, who have found mRNA traces in transplant models [41]. *In vivo* traces of Comirnaty and Spikevax spike have been detected before in different cells and tissues [79]. Previously, mRNA could be detected in the human placenta using RT-PCR [80]. Comparable to a previous study with two analysed placentas [44], mRNA traces of the modRNA vaccines were observed *in vivo* in the placenta in two of nine spike-protein positive individual samples without any known significant health consequences in mother and child. Other than Prahl and coworkers who have excluded traces of the vaccine in 20 placentas by Western blot and RT-PCR [81] we had a few positive samples; we assume that several hits must coincide for transmission or uptake of the mRNA to occur in the placenta. Due to the exploratory approach with only nine, albeit spike protein positive samples, generalizing interpretation of our RNAscope results should be considered cautiously.

Following maternal COVID-19, the occurrence of placentitis has also been described, which consists of trophoblast necrosis, perivillous fibrin deposition and chronic histiocytic intervillositis [82]. This histopathologic triad occurrence was associated with a negative pregnancy outcome [83, 84]. In various studies the rate of SARS-CoV-2 placentitis varied to a maximum of one fifth or one quarter [83, 85]. However, there is also published literature describing the absence of any specific histopathological alterations (for review of both [86]). Correspondingly, we did not see this either.

The limitations of our study were a relatively small number of unvaccinated women with COVID-19, which was due to the rather low SARS-CoV-2 infection incidence at the beginning of the pandemic in our region. Furthermore, the study population only consisted of women who were admitted to the hospital to give relatively uncomplicated child births since most of women with an expected premature or complicated birth did not agree to participate in the trial. Women with miscarriages were not included.

## 5. Conclusions

In our trial, spike protein of SARS-CoV-2 could be detected in placental Hofbauer and trophoblast cells as well as in villous capillary endothelia after infection and/or vaccination indicating a possible transplacental effect or uptake or both. Further, we have found residues of the vaccine RNA via RNAScope technique but no viral RNA in two individual spike protein samples. No correlation was seen between spike protein or modRNA detection in the placentas with the medical outcome of mother and child due to our inclusion criteria and the small cohort. Therefore, we would like to encourage to reproduce this investigation on a larger collective.

## Supporting information

**S1 Table. Medical data with regard to COVID-19, vaccination as well as immunochemistry for 31 placenta samples with positive spike protein results.**
(XLSX)

**S2 Table. Table with the row data according to the FAIR principles.**
(XLSX)

## Acknowledgments

We thank our midwifes for their support with participant and material recruitment every day and night in the delivery rooms, Michaela Kapp for her excellent technical assistance and Angela Mayr-Isenberg for her careful proofreading.

## Author contributions

**Conceptualization:** Catharina Bartmann, Ulrike Kämmerer.

**Data curation:** Michael Schwab, Monika Rehn.

**Formal analysis:** Catharina Bartmann, Ulrike Kämmerer.

**Methodology:** Catharina Bartmann, Vanessa Schmidt, Michael Mörz, Ulrike Kämmerer.

**Resources:** Achim Wöckel.

**Software:** Achim Wöckel.

**Supervision:** Vanessa Schmidt, Michael Mörz.

**Validation:** Vanessa Schmidt, Michael Mörz.

**Writing – original draft:** Catharina Bartmann, Ulrike Kämmerer.

**Writing – review & editing:** Catharina Bartmann, Vanessa Schmidt, Michael Mörz, Michael Schwab, Monika Rehn, Bettina Blau-Schneider, Achim Wöckel, Ulrike Kämmerer.

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
