## [Decision Letter · Decision Letter 0]

21 Oct 2025

PONE-D-25-37690Detection of Spike protein in term placentas of COVID-19 vaccinated and/or SARS-CoV-2 infected womenPLOS ONE

Dear Dr. Bartmann,

Thank you for submitting your manuscript to PLOS ONE. After careful consideration, we feel that it has merit but does not fully meet PLOS ONE’s publication criteria as it currently stands. Therefore, we invite you to submit a revised version of the manuscript that addresses the points raised during the review process.

We look forward to receiving your revised manuscript.

Kind regards,

Jayonta Bhattacharjee

Academic Editor

PLOS ONE

Journal Requirements:

Reviewer's Responses to Questions

**Comments to the Author**

1. Is the manuscript technically sound, and do the data support the conclusions?

Reviewer #1: No

Reviewer #2: Yes

2. Has the statistical analysis been performed appropriately and rigorously? 

Reviewer #1: Yes

Reviewer #2: Yes

3. Have the authors made all data underlying the findings in their manuscript fully available?

Reviewer #1: Yes

Reviewer #2: Yes

4. Is the manuscript presented in an intelligible fashion and written in standard English?

Reviewer #1: No

Reviewer #2: Yes

5. Review Comments to the Author

Reviewer #1: In this manuscript, the authors describe a cohort of pregnant people who experienced COVID-19 infection during the SARS-CoV-2 global pandemic who had received vaccination (or no vaccination) in Germany. The analysis includes detailed histological examination of placental tissues for various proteins associated with SARS-CoV-2. The authors describe their findings systematically, and test whether obstetric and neonatal outcomes were affected by SARS-CoV-2 infection or vaccination.

A major pitfall of the current investigation is the lack of co-staining with markers of Hofbauer cells for example to show localization of SARS-CoV-2-related proteins. I would say that this is necessary for publication.

There are many instances of grammatical errors, spelling mistakes, and general misuse of punctuation that requires immediate correction. The authors are encouraged to more finely review the manuscript before initial submission to facilitate efficient review by reviewers. This manuscript reads as though the authors were not careful in their preparation of this manuscript for submission to PLOS ONE.

Abstract:

• The abstract is well-written and concise. No amendments are recommended.

Introduction:

• Lines 79-81: This sentence does not make grammatical sense. The authors are encouraged to review the manuscript carefully for grammatical errors and inconsistencies.

• Line 83: Be careful with use of tenses. For example, here it should be “neonatal morbidities were reported in 2021”. There are multiple instances in this manuscript where past-tense should be used and not present-tense.

• Line 105: The comma is not needed here.

• Line 116: Comma after “we were interested” is not necessary, please remove.

Methods:

• Line 130: October not Oktober.

• Table 1: 2nd and 3rd spelled incorrectly.

• Line 150: Should be Women (plural), not Woman (singular).

• Please include dilutions used of all antibodies (perhaps in a table in the supplementary data).

Results:

• Starting at Line 227: Could any of these numbers be expressed as percentages so the reader can appreciate the proportions affected?

• Figures: Please capitalize the start of every individual text label (i.e., “Number of Patients”, “Symptoms” – both axis labels.

• Line 270: Correct to leukocytes not leucocytes.

• Table 4: Correct to negative, not negativ; Please clean up the figure and make sure things are capitalized where needed, and that abbreviations are listed in the footnotes. There are many instances of spelling mistakes that need to be corrected.

• Line 278: “was” not “war”.

• Line 281: hints? Should this be “hits”?

• Line 283: what is “trimenon”?

• Figures 3&4: Please add scale bars to your images. Further, if you are going to claim SARS-CoV-2 proteins in Hofbauer cells for example, you would absolutely need co-staining with a marker for Hofbauer cells like CD68 or CD163. I would say that without some distinct IHC staining of differing cell types this manuscript loses a lot of its strength. I would request co-staining likely by IF would be sufficient. I likely would not accept a paper for publication without this confirmation.

Discussion:

• Line 291: What is meant by “assumed emergency situation”? A conservative estimate of COVID-19 deaths globally puts it at 7 million deaths. More accurate estimates by the WHO the Lancet put the deaths at 16-18 million globally. If this isn’t an emergency, I don’t know what is.

• Line 300-301: This sentence is not understandable.

• You have listed a major limitation of your study to exclude those with complicated births, and I would like to know why there was a conscious decision to include cases where COVID-19 may have caused harm to mother and fetus/neonate.

• Line 349: What is trasplacentar?

Reviewer #2: Dear Authors:

This study investigated whether SARS-CoV-2 spike protein or vaccine-derived mRNA can be detected in term placentas. The cohort included 106 women who delivered at the University Hospital of Würzburg between November 2020 and October 2022. The strengths include:

1. Novelty: First reported detection of vaccine-derived mRNA traces in placental tissue in vivo.

2. Comprehensive Approach: Combined immunohistochemistry and RNAscope, increasing reliability of detection.

In general, this is a well-written manuscript, and I have no further comments.

6. PLOS authors have the option to publish the peer review history of their article (what does this mean? ). If published, this will include your full peer review and any attached files.

**Do you want your identity to be public for this peer review?** For information about this choice, including consent withdrawal, please see our Privacy Policy .

Reviewer #1: No

Reviewer #2: **Yes:** CHIEN-YU CHENG

---

## [Author Response · Author response to Decision Letter 1]

26 Dec 2025

Reviewer #1: In this manuscript, the authors describe a cohort of pregnant people who experienced COVID-19 infection during the SARS-CoV-2 global pandemic who had received vaccination (or no vaccination) in Germany. The analysis includes detailed histological examination of placental tissues for various proteins associated with SARS-CoV-2. The authors describe their findings systematically, and test whether obstetric and neonatal outcomes were affected by SARS-CoV-2 infection or vaccination.

Thank you.

A major pitfall of the current investigation is the lack of co-staining with markers of Hofbauer cells for example to show localization of SARS-CoV-2-related proteins. I would say that this is necessary for publication.

Thank you. We have addressed the major pitfall. CD68 is a well-established and widely used marker for the identification of Hofbauer cells. Therefore, we have performed double immunostaining with anti-CD68 and anti-spike antibodies, both of which stained Hofbauer cells. Accordingly, we have included an additional figure (new Figure 5, previous Figure 5 is now Figure 6) to further illustrate our findings.

There are many instances of grammatical errors, spelling mistakes, and general misuse of punctuation that requires immediate correction. The authors are encouraged to more finely review the manuscript before initial submission to facilitate efficient review by reviewers. This manuscript reads as though the authors were not careful in their preparation of this manuscript for submission to PLOS ONE.

Thank you for pointing this out. We have read the text carefully. Further, the manuscript has been proofread and corrected by a certified translator.

Abstract:

• The abstract is well-written and concise. No amendments are recommended.

Ok.

Introduction:

• Lines 79-81: This sentence does not make grammatical sense. Corrected

The authors are encouraged to review the manuscript carefully for grammatical errors and inconsistencies. Done

• Line 83: Be careful with use of tenses. For example, here it should be “neonatal morbidities were reported in 2021”. There are multiple instances in this manuscript where past-tense should be used and not present-tense. Done

• Line 105: The comma is not needed here. Done

• Line 116: Comma after “we were interested” is not necessary, please remove. Done

Methods:

• Line 130: October not Oktober. Done

• Table 1: 2nd and 3rd spelled incorrectly. Done

• Line 150: Should be Women (plural), not Woman (singular). Done

• Please include dilutions used of all antibodies (perhaps in a table in the supplementary data).

We have created an additional table (new Table 3) that contains the primary antibodies for immunohistochemistry, including the dilutions.

Results:

• Starting at Line 227: Could any of these numbers be expressed as percentages so the reader can appreciate the proportions affected? Done

• Figures: Please capitalize the start of every individual text label (i.e., “Number of Patients”, “Symptoms” – both axis labels. Done

• Line 270: Correct to leukocytes not leucocytes. Done

• Table 4: Correct to negative, not negativ Done;

• Please clean up the figure and make sure things are capitalized where needed, and that abbreviations are listed in the footnotes. There are many instances of spelling mistakes that need to be corrected. Thank you very much. We have improved the previous Figure 5 (now Figure 6).

• Line 278: “was” not “war”. Corrected

• Line 281: hints? Should this be “hits”? No, we mean hint in the meaning of “point to”. Now we have used the term “proof” since the RNA-detection indeed is a proof of the presence of the RNA in question, we have rewritten this sentence.

• Line 283: what is “trimenon”? trimenon is trimester, we have replaced it now with trimester

• Figures 3&4:

Please add scale bars to your images. Done.

Further, if you are going to claim SARS-CoV-2 proteins in Hofbauer cells for example, you would absolutely need co-staining with a marker for Hofbauer cells like CD68 or CD163. I would say that without some distinct IHC staining of differing cell types this manuscript loses a lot of its strength. I would request co-staining likely by IF would be sufficient. I likely would not accept a paper for publication without this confirmation.

Thank you for the comment. We have performed double staining and added a typical example as a new Figure 5.

Discussion:

• Line 291: What is meant by “assumed emergency situation”? A conservative estimate of COVID-19 deaths globally puts it at 7 million deaths. More accurate estimates by the WHO the Lancet put the deaths at 16-18 million globally. If this isn’t an emergency, I don’t know what is.

Well that’s still a matter of debate since overall mortality was not detected in the national statistics databases for Germany ( https://www.destatis.de/DE/Themen/Gesellschaft-Umwelt/Bevoelkerung/Sterbefaelle-Lebenserwartung/_inhalt.html#sprg229094). Since our study groups were recruited in Würzburg/Germany, we refer to those data in our interpretation. According to a very new publication (Kuhbandner C, Reitzner M. 2025 Regional patterns of excess mortality in Germany during the COVID-19 pandemic: a state-level analysis. R. Soc. Open Sci. 12: 250790. https://doi.org/10.1098/rsos.250790rsos.250790.pdf) there was only a moderate excess mortality – comparable to years with a severe influenza season, such as the flu season 2017/2018. However, we have deleted the phrase and replaced it by “following European guidelines”, to avoid this discussion, which is not a matter of our publication.

• Line 300-301: This sentence is not understandable.

We have improved the sentence adding “group compared to the general German population”.

• You have listed a major limitation of your study to exclude those with complicated births, and I would like to know why there was a conscious decision to include cases where COVID-19 may have caused harm to mother and fetus/neonate.

All women were asked during the recruitment period whether they would like to participate in the study. We did not exclude any premature births. However, we always ensured that the medical safety of the women and their children took priority. Overall, only a few women with premature births participated in the study, meaning that the average premature birth rate in Germany is underrepresented. The sentence “Women with miscarriages and early premature births were not part of the study population.” is misunderstanding, therefore we have improved it.

• Line 349: What is trasplacentar? This was a typo, it should mean “transplacental”. We have corrected this.

Reviewer #2: Dear Authors:

This study investigated whether SARS-CoV-2 spike protein or vaccine-derived mRNA can be detected in term placentas. The cohort included 106 women who delivered at the University Hospital of Würzburg between November 2020 and October 2022. The strengths include:

1. Novelty: First reported detection of vaccine-derived mRNA traces in placental tissue in vivo.

2. Comprehensive Approach: Combined immunohistochemistry and RNAscope, increasing reliability of detection.

In general, this is a well-written manuscript, and I have no further comments.

Thank you.

---

## [Decision Letter · Decision Letter 1]

13 Jan 2026

PONE-D-25-37690R1Detection of Spike protein in term placentas of COVID-19 vaccinated and/or SARS-CoV-2 infected womenPLOS One

Dear Dr.Catharina Bartman,

Thank you for submitting your manuscript to PLOS ONE. After careful consideration, we feel that it has merit but does not fully meet PLOS ONE’s publication criteria as it currently stands. Therefore, we invite you to submit a revised version of the manuscript that addresses the points raised during the review process.

We look forward to receiving your revised manuscript.

Kind regards,

Moises Leon Juarez

Academic Editor

PLOS One

Journal Requirements:

Additional Editor Comments (if provided):

After careful evaluation of the manuscript, I consider that it presents original observational research addressing a timely and sensitive topic, namely the detection of SARS-CoV-2 spike protein and vaccine-related RNA signals in term placental tissue following maternal infection and/or vaccination. The study is based on a well-defined clinical cohort, and the authors apply appropriate histological and molecular techniques that are suitable for the descriptive aims of the work.

From a technical perspective, the experimental approaches are conducted to an adequate standard and are described in sufficient detail to allow reproducibility. The immunohistochemical analyses include appropriate controls, and the use of double immunostaining to identify Hofbauer cells strengthens the interpretation of cellular localization. The exploratory application of RNAscope provides an additional layer of supportive evidence, although it should be interpreted cautiously. Statistical analyses are appropriate for the study design, and ethical approval, informed consent, and data availability are clearly documented and meet journal requirements.

However, several points require editorial clarification and refinement, particularly in light of the sensitivity of the topic. The study is observational and largely descriptive, and the data do not support functional, mechanistic, or causal inferences regarding viral replication, biological activity, or fetal clinical impact. It is therefore essential that the Discussion explicitly reflects these limitations and avoids over-interpretation.

Specifically, the authors should clarify that immunohistochemical detection of spike protein does not distinguish between locally synthesized protein and protein taken up from the circulation, and that in macrophage-lineage cells such as Hofbauer cells, the observed signal may reflect phagocytosis or antigen uptake rather than active infection. Similarly, RNAscope results should be clearly contextualized as exploratory, performed on a limited number of samples, and not indicative of replication competence or biological activity. The absence of a clearly presented negative control for RNAscope should also be addressed.

In addition, discrepancies between spike and nucleocapsid immunoreactivity should be discussed, particularly given prior reports showing more comparable detection of both antigens. Claims of novelty regarding the detection of vaccine-derived RNA in the placenta must be revised, as similar findings have already been reported in the literature. The authors should clearly articulate what is novel about the present study relative to these prior reports.

Minor issues, such as clarification of symbols (circles and arrows) in Figure 6, should also be corrected.

Reviewers' comments:

Reviewer's Responses to Questions

**Comments to the Author**

1. If the authors have adequately addressed your comments raised in a previous round of review and you feel that this manuscript is now acceptable for publication, you may indicate that here to bypass the “Comments to the Author” section, enter your conflict of interest statement in the “Confidential to Editor” section, and submit your "Accept" recommendation.

Reviewer #3: (No Response)

Reviewer #4: (No Response)

2. Is the manuscript technically sound, and do the data support the conclusions?

Reviewer #3: Yes

Reviewer #4: No

3. Has the statistical analysis been performed appropriately and rigorously? 

Reviewer #3: Yes

Reviewer #4: N/A

4. Have the authors made all data underlying the findings in their manuscript fully available?

Reviewer #3: Yes

Reviewer #4: No

5. Is the manuscript presented in an intelligible fashion and written in standard English?

Reviewer #3: Yes

Reviewer #4: Yes

6. Review Comments to the Author

Reviewer #3: After reviewing the manuscript, I find that it presents original observational research addressing a timely and sensitive topic the detection of SARS-CoV-2 spike protein and vaccine-related RNA signals in term placental tissue following maternal infection and/or vaccination. The study is based on a well-defined clinical cohort and applies appropriate histological and molecular techniques to address its descriptive aims.

From a technical standpoint, the experiments are conducted to an adequate standard and are described in sufficient detail to allow reproducibility. The immunohistochemical analyses are supported by appropriate controls, and the addition of double staining to identify Hofbauer cells strengthens the interpretation regarding cellular localization. The exploratory use of RNAscope provides an additional layer of evidence, which is presented as supportive rather than definitive.

The statistical analyses are appropriate for the study design, and the conclusions are generally aligned with the data. Ethical approval, informed consent, and data availability are clearly documented and meet the journal’s requirements.

Given the sensitive nature of the topic, I would encourage the authors to add a small amount of additional interpretative caution in the discussion. In particular, it would be useful to state explicitly that immunohistochemical detection of spike protein does not distinguish between locally produced protein and protein that has been taken up from the circulation, and that in macrophage-lineage cells such as Hofbauer cells the signal may also reflect phagocytosis or antigen uptake rather than active infection in order to avoid over-interpretation. Along similar lines, a brief sentence noting that RNAscope analyses were exploratory and performed on a limited number of selected samples would better contextualize those results, and it would also be helpful to explicitly state that detection does not imply biological activity, replication competence, or fetal clinical impact.

Reviewer #4: I consider that Dr. Bartman and collaborators have a treasure trove of biological samples (placentas); however, the data they present leaves much to be desired, because in the immunohistochemistry, of three placentas positive for Nucleocapsid, if I understand correctly there is a signal for both viral antigens (Spike and N); however, there is a lot of positive signal for the Spike protein and very little signal for the Nucleocapsid protein, which contrasts with several articles, where a similar signal for both antigens has been reported. (They could discuss this part). Another core aspect of their work is their RNA scope, where they do not show a negative control, as apparently very little signal is observed in the positive samples, which leads me to think it could be a false positive. Finally, the authors mention that their work is novel because it is the first time traces of vaccine RNA in the placenta are reported, which is false, as it has already been reported by Xinhua. Lin, and published in JUNE of 2024, in American Journal of Obstetrics & Gynecology e115 (Transplacental transmission of the COVID-19 vaccine messenger RNA: evidence from placental, maternal, and cord blood analyses postvaccination).

In this sense, what would be new about the work???

By other hand Morgenstern Milana en el 2025, examineted the persistence of synthetic mRNA from the COVID-19 vaccine Comirnaty in placenta of vaccined individuals. And Vaccine mRNA was detected in most samples from vaccinated individuals including placenta tissue (DOI: 10.29011/2574-7754.102428).

Finally, in Figure 6, they do not mention what the circles or the arrows represent.

7. PLOS authors have the option to publish the peer review history of their article (what does this mean? ). If published, this will include your full peer review and any attached files.

**Do you want your identity to be public for this peer review?** For information about this choice, including consent withdrawal, please see our Privacy Policy .

Reviewer #3: No

Reviewer #4: No

---

## [Author Response · Author response to Decision Letter 2]

12 Feb 2026

Reviewer #3:

After reviewing the manuscript, I find that it presents original observational research addressing a timely and sensitive topic the detection of SARS-CoV-2 spike protein and vaccine-related RNA signals in term placental tissue following maternal infection and/or vaccination. The study is based on a well-defined clinical cohort and applies appropriate histological and molecular techniques to address its descriptive aims.

Thank you!

From a technical standpoint, the experiments are conducted to an adequate standard and are described in sufficient detail to allow reproducibility. The immunohistochemical analyses are supported by appropriate controls, and the addition of double staining to identify Hofbauer cells strengthens the interpretation regarding cellular localization. The exploratory use of RNAscope provides an additional layer of evidence, which is presented as supportive rather than definitive.

The statistical analyses are appropriate for the study design, and the conclusions are generally aligned with the data. Ethical approval, informed consent, and data availability are clearly documented and meet the journal’s requirements.

Thank you!

Given the sensitive nature of the topic, I would encourage the authors to add a small amount of additional interpretative caution in the discussion. In particular, it would be useful to state explicitly that immunohistochemical detection of spike protein does not distinguish between locally produced protein and protein that has been taken up from the circulation, and that in macrophage-lineage cells such as Hofbauer cells the signal may also reflect phagocytosis or antigen uptake rather than active infection in order to avoid over-interpretation.

Thank you; we have clarified this point more explicitly in the revised version of our manuscript.

Along similar lines, a brief sentence noting that RNAscope analyses were exploratory and performed on a limited number of selected samples would better contextualize those results, and it would also be helpful to explicitly state that detection does not imply biological activity, replication competence, or fetal clinical impact.

Due to the exploratory approach with only nine samples, the interpretation of our results should be considered cautiously. We have made it very clear in the manuscript to prevent misunderstandings. Reviewer #4:

I consider that Dr. Bartman and collaborators have a treasure trove of biological samples (placentas); however, the data they present leaves much to be desired, because in the immunohistochemistry, of three placentas positive for Nucleocapsid, if I understand correctly there is a signal for both viral antigens (Spike and N); however, there is a lot of positive signal for the Spike protein and very little signal for the Nucleocapsid protein, which contrasts with several articles, where a similar signal for both antigens has been reported. (They could discuss this part).

Thank you for pointing that out. When staining with spike and nucleocapsid antibodies, it is important to differentiate between the protein expression by vaccination effect, infection and the lingering effects of an infection. This is already part of our ‘Discussion’ section. For clearer understanding, we have discussed this point in detail.

Another core aspect of their work is their RNA scope, where they do not show a negative control, as apparently very little signal is observed in the positive samples, which leads me to think it could be a false positive.

Thank you for your important advice. Negative controls were already demonstrated in Figure 6. We have added the information of negative controls to the main manuscript.

Figure A, B: Placental specimen of a woman vaccinated by Comirnaty before and during pregnancy and with COVID-19 in the 36th week of pregnancy.

Figure 6A showed the positive result (BNT162b2-C1), Figure 6B the negative control (sensors V-nCOV2019-S Wuhan).

Figure C, D: Placental specimen of a woman vaccinated by Spikevax before pregnancy.

Figure 6C showed the positive result (S-encoding-mRNA-1273), Figure 6B the negative control (sensors V-nCOV2019-S Wuhan).

Finally, the authors mention that their work is novel because it is the first time traces of vaccine RNA in the placenta are reported, which is false, as it has already been reported by Xinhua. Lin, and published in JUNE of 2024, in American Journal of Obstetrics & Gynecology e115 (Transplacental transmission of the COVID-19 vaccine messenger RNA: evidence from placental, maternal, and cord blood analyses postvaccination). In this sense, what would be new about the work???

Many thanks for your comment. We have included both papers in our discussion and corrected the facts.

Finally, in Figure 6, they do not mention what the circles or the arrows represent.

Thank you for your advice. We have added it.

---

## [Editor Report · Decision Letter 2]

18 Feb 2026

Detection of Spike protein in term placentas of COVID-19 vaccinated and/or SARS-CoV-2 infected women

PONE-D-25-37690R2

Dear Dr. Bartmann,

We’re pleased to inform you that your manuscript has been judged scientifically suitable for publication and will be formally accepted for publication once it meets all outstanding technical requirements.

Kind regards,

Moises Leon Juarez

Academic Editor

PLOS One

Additional Editor Comments (optional):

In accordance with the comments and suggestions provided by the reviewers regarding the manuscript entitled “Detection of Spike Protein in Term Placentas of COVID-19 Vaccinated and/or SARS-CoV-2 Infected Women,” the authors have implemented substantial revisions addressing the points raised. These modifications have significantly improved the clarity, coherence, and organization of the results section.

In light of these revisions, I consider that the manuscript has been strengthened and now meets the standards required for publication in PLOS ONE. Therefore, I recommend its acceptance for publication.
---

## [Editor Report · Acceptance letter]

PONE-D-25-37690R2

PLOS One

Dear Dr. Bartmann,

I'm pleased to inform you that your manuscript has been deemed suitable for publication in PLOS One. Congratulations! Your manuscript is now being handed over to our production team.

Kind regards,

on behalf of

Dr. Moises Leon Juarez

Academic Editor

PLOS One